# Somatostatin, a Presynaptic Modulator of Glutamatergic Signal in the Central Nervous System

**DOI:** 10.3390/ijms22115864

**Published:** 2021-05-30

**Authors:** Anna Pittaluga, Alessandra Roggeri, Giulia Vallarino, Guendalina Olivero

**Affiliations:** 1Department of Pharmacy, School of Medical and Pharmaceutical Sciences, University of Genova, 16148 Genova, Italy; alessandra.roggeri@libero.it (A.R.); giulia.vallarino94@gmail.com (G.V.); olivero@difar.unige.it (G.O.); 2IRCCS S. Martin Hospital, 16145 Genova, Italy; 33Rs Center, University of Genova, 16145 Genova, Italy

**Keywords:** somatostatin, glutamate, noradrenaline, metamodulation, volume diffusion, NMDA receptors, sst2 receptors, sst5 receptors

## Abstract

Somatostatin is widely diffused in the central nervous system, where it participates to control the efficiency of synaptic transmission. This peptide mainly colocalizes with GABA, in inhibitory, GABA-containing interneurons from which it is actively released in a Ca^2+^ dependent manner upon application of depolarizing stimuli. Once released in the synaptic cleft, somatostatin acts locally, or it diffuses in the extracellular space through “volume diffusion”, a mechanism(s) of distribution which mainly operates in the cerebrospinal fluid and that assures the progression of neuronal signalling from signal-secreting sender structures towards receptor-expressing targeted neurons located extrasynaptically, in a non-synaptic, inter-neuronal form of communication. Somatostatin controls the efficiency of central glutamate transmission by either modulating presynaptically the glutamate exocytosis or by metamodulating the activity of glutamate receptors colocalized and functionally coupled with somatostatin receptors in selected subpopulations of nerve terminals. Deciphering the role of somatostatin in the mechanisms of “volume diffusion” and in the “receptor-receptor interaction” unveils new perspectives in the central role of this fine tuner of synaptic strength, paving the road to new therapeutic approaches for the cure of central disorders.

## 1. The Somatostatinergic System in the Central Nervous System

The cyclic tetradecapeptide somatostatin was initially identified as a hypothalamic peptide that regulates growth hormone secretion from cells of the anterior pituitary gland [1]. Soon after, data were provided showing that somatostatin is widely expressed in the central nervous system (CNS), where it plays a main role as regulator of neuronal functions [2]. Here, the peptide exists in two main biologically active forms, the somatostatin-14 (SRIF-14), which is the peptide firstly discovered and the amino-terminal extended form, we refer to as the somatostatin-28 (SRIF-28). In mammals, somatostatin originates from a single gene that encodes for preprosomatostatin (116 amino acids) which is cleaved into prosomatostatin (96 amino acids) from which SRIF-14 and SRIF-28 originate [3,4,5,6].

Somatostatin mediates its actions through a family of G protein-coupled receptors (GPCRs) having a structural distribution in cell membranes. These receptors are typified by seven transmembrane domains that couple either to inhibitory or facilitatory G proteins [7]. Five genes encoding the distinct somatostatin receptor subtypes, we refer to as sst1-sst5 receptors, have been decoded in humans and other species. Among these structures, the carboxy-terminal tail of the sst2 receptor undergoes alternative splicing further yielding to other two isoforms, namely the sst2A and sst2B receptors [8,9,10]. The two natural active forms of somatostatin (namely SRIF-14 and SRIF-28) have comparable affinities towards the different receptor subtypes and cannot be used to pharmacologically discriminate them [11].

In neurons, the members of the somatostatinergic receptor family couple inhibitory G proteins (Gi/Go), whose α subunit negatively controls the adenylyl cyclase (AC)-mediated pathway. The activation of these receptors leads to reduced production of cyclic adenosine monophosphate (cAMP) and consequent hypoactivation of protein kinase A (PKA)-mediated pathway(s), including those controlling the opening of the voltage-operated calcium channels (VOCCs). The main consequence of the translocation of the α subunit is a reduced influx of Ca^2+^ ions and consequently only a partial activation of the associated intraterminal events. Concomitantly, also the βγ subunits are mobilized and translocated to activate inward-rectifier K^+^ channels. The signal originated from this multistep-intraterminal enzymatic pathway is by one side the hyperpolarization of the neuronal plasma membranes due to the increased K^+^ ions influx and, on the other side, the partial inactivation of the VOCCs and the reduction of the AC activity. These concomitant events converge to silence most of the cellular-mediated processes. 

It is, however, worth stressing that some of the sst receptor subtypes (namely the sst2, sst3, sst5 receptors) also link facilitatory G proteins which positively couple the protein phospholipase C (PLC). In this case, the activation of the receptors favours the translocation of PLC to the neuronal membranes, where the enzyme triggers the hydrolysis of membrane phosphoinositides to produce inositol trisphosphate (IP_3_) and diacylglycerol (DAG), which in turn activate intraterminal protein kinase C (PKC)-dependent events and related phosphorylative signalling (Figure 1) [7,9,10]. Notably, this pathway also leads to the phosphorylation of cytosolic tyrosin kinases (src) that in turn control cellular functions (i.e., by modulating the activity of other receptors, as well as by triggering apoptotic events in selected subpopulations of cells, including the tumoral ones) [7,12].

Starting from the first evidence supporting the existence of sst receptor subtypes, efforts were made to classify them by a pharmacological point of view and to identify lead compounds that could unveil receptor-directed intervention for the cure of pathological conditions. The pharmacological classification of the sst receptors was first based on the affinity of the receptor for selective peptidergic somatostatin-like ligands, particularly octreotride and seglitide [9,11,13]. Starting from the late 1990s, however, non-peptide ligands typified by huge selectivity towards selected receptor subtypes also became available [14,15]. These non-peptidergic compounds had a great impact on the subclassification of the sst receptors, as well as on the definition of their physio-pathological roles, since they permitted to bypass some limitations due to the peptidergic nature of the available ligands, including the unfavourable pharmacokinetic profile in in vivo studies, as well as the unpredictable distribution in in vitro models typified by a high structural complexity (i.e., brain tissue slices). 

As far as the neuronal compartment in the CNS is concerned, the pharmacological profile and the main structural features of the receptor subtypes led to classify them into two main groups: the somatostatin release-inhibiting factor group 1 (SRIF1), including the sst2, 3 and 5 receptors and the SRIF2 group, consisting of the sst1 and sst4 receptor subtypes with a prevalent postsynaptic localization [16,17]. Starting from these findings, the pharmacological efforts to draw new, even more selective ligands grew up and nowadays new classes of peptidergic and non-peptidergic molecules are available for both clinical and preclinical studies [5,9]. 

In the CNS, the somatostatinergic system is involved in several cellular events subserving physiological processes related either to the endocrine or the exocrine control of hormone release, as well as to molecular processes underlying cognition, sleep, and motor coordination. Somatostatin also participates to the onset and the development of pathological events involving cognitive and motor impairments and even neurodegenerative disorders such as Alzheimer’s disease, epilepsy, and tumours. All these aspects have been already discussed in dedicated reviews [5,7,13,16] and will be not recapitulated in this work. 

## 2. Somatostatin in the Central Nervous System: Synaptic Release and Volume Diffusion 

In the CNS, peptides are produced at the endoplasmic reticulum, far from the synaptic processes and stored in dense core vesicles that are then transported along the axonal processes to the synaptic varicosities and nerve terminals specialized for neurotransmitter exocytosis. Data available in the literature suggest that these dense core vesicles preferentially do not release their content into the synaptic active zone, but rather nearby these structures, perisynaptically (Figure 2) ([18] and references therein). In most cases, the vesicles also contain non-peptidergic neurotransmitters which are co-stored and co-released with peptides upon depolarization of axonal terminals [19,20,21]. As far as the somatostatin is concerned, somatostatinergic-containing dense core vesicles are mostly present in interneuronal networks innervating the septum, the striatum, the hippocampus and the cortex of mammals [22,23,24,25]. The neocortex and the hippocampus represent the brain regions showing the most diffuse somatostatinergic immunopositivity [26]. The main features of the somatostatin-expressing interneurons in the cerebral cortex and their role in controlling cortical circuits in physio-pathological conditions have been recently described by Riedemann [27] and will be not further discussed in the present review.

Somatostatin is actively released in a Ca^2+^ dependent manner from neurons [28] and from isolated nerve terminals (we refer to as synaptosomes) upon their exposure to classic depolarizing stimuli (9 to 50 mM KCl-enriched solution or 1 to 50 µM veratrine) [29]. The efficiency of somatostatin exocytosis relies on a cascade of events that strictly depends on external Na^+^, as suggested by the fact that tetrodotoxin, which is known to hamper the opening of voltage-dependent Na^+^ channels, largely reduced the peptide overflow [29]. 

The study of the role of the Na^+^ ions in controlling the release of somatostatin, however, unveiled a more complicated scenario than expected. Although the secretion of somatostatin from cortical nerve endings in both resting and depolarized conditions was strictly dependent on the membrane Na^+^ gradient, the correlation linking the somatostatin release and the [Na^+^]_out_ shifts from positive to negative depending on the nature of the stimulus applied. 

The [Na^+^]_out_ was found to be positively associated with somatostatin exocytosis elicited by a mild depolarizing stimulus, but the correlation became negative when analysing the release of the peptide in resting conditions. In the latter case, lowering the extracellular concentration of Na^+^ (but not increasing it) unveiled a Ca^2+^-insensitive, non-exocytotic release of somatostatin. To the best of our knowledge, the mechanism(s) accounting for the low Na^+^-induced releasing activity remains so far unknown, also because the involvement of an uptake carrier working in the reverse mode (as proposed for classic neurotransmitters) seems unlike, mainly because of the lack of reuptake system(s) specific for somatostatin in nerve terminals. Interestingly, beside Na^+^ ions, also protons largely influence the efficiency of peptide outflow. Alkalinization (but not acidification) of the external milieu was shown to increase somatostatin release through a cascade of events leading to a conventional exocytosis but also to a concomitant outflow that is independent from external Ca^2+^ ions [29]. 

It is worth reminding that all the above-described results were obtained in release studies carried out with the technique of “superfusion of a thin layer of synaptosomes”. This method, first proposed by Raiteri and colleagues in 1974 [30], represents an approach of choice to quantify the release of endogenous peptides/neurotransmitters from isolated synaptosomes and to investigate the molecular mechanisms underlying these events. The main feature of the technique is the rapid removal of any endogenous compounds/enzymes released by the superfused particles, which limits indirect effects, including the enzymatic degradation of the peptides [29]. This characteristic makes this experimental approach particularly appropriate to quantify the release of somatostatin in both resting and stimulated conditions, allowing also to correlate the efficiency of the peptide exocytosis to the intensity of the stimulus applied [31]. 

The functional evidence so far briefly summarized support the conclusion that somatostatin is released from nerve terminals by different releasing stimuli, including conditions that mimic pathological situations (i.e., the alkalinization of the extracellular space), providing an experimental model for studying the mechanisms of adaptation/maladaptation of somatostatin release in the course of central disorder(s). 

An important feature of neuropeptides is that they often colocalize with classic neurotransmitters. As already introduced, somatostatin is preferentially expressed in inhibitory, GABA-containing interneurons. The highest level of colocalization between GABA and somatostatin is observed in hippocampal and cortical interneurons [27,32] that efficiently release the two neuromodulators in stimulated conditions in a calcium-dependent manner [33].

In a classic view, the neurotransmitters/peptides that are released upon the application of a depolarizing stimulus diffuse in the synaptic cleft to reach retrogradely receptors that control the functions of the neuronal processes they are released from. Concomitantly, the neurotransmitters/peptides can also activate receptors located postsynaptically and/or on surrounding cells, to propagate the chemical signalling. In other words, at chemical synapses, neurotransmitters /peptides can exert either autocrine or paracrine effects controlling the strength and the efficiency of the synaptic connection (Figure 2). 

In this regard, GABA released from interneurons retrogradely controls its own exocytosis, but also modulates the outflow of the colocalized somatostatin [34,35]. Quite interestingly, the results in literature demonstrated that both the presynaptic autoreceptors controlling GABA exocytosis and the GABA heteroreceptors modulating somatostatin release belong to the GABA_B_ receptor subtype but with different pharmacological profiles [35]. Whether the auto- and the heteroreceptors colocalize and they share a common presynaptic pathway(s) was not so far investigated. Interestingly, presynaptic GABA_B_ heteroreceptors inhibiting somatostatin exocytosis were also shown to exist in synaptosomes isolated from human cortical specimens removed during neurosurgery to reach deep-located tumours. The GABA_B_ heteroreceptors in human terminals displayed a pharmacological profile comparable to that of the receptors in rat cortical synaptosomes [34].

As far as somatostatin is concerned, the data in the literature unveiled a more complex scenario. In 1998, Dournaud and colleagues [36] investigated the distribution of the sst2 receptors in selected regions of the CNS and correlated this parameter to the distribution of the somatostatin-containing terminals. The study was limited to the sst2 receptors, which account for a relevant percentage of the somatostatin receptors in the CNS. It focused on the relative distribution of the somatostatin-containing axons and of the sst2 receptors expressing neurons, providing an important picture of the central somatostatinergic innervation. The authors described two main patterns of immunostaining, the first showing a clear somato-dendritic distribution, the second typified by a diffuse immunopositivity within the tissue. Interestingly, the sst2 receptor labelling in the somato-dendritic regions was differently scattered in the neuronal compartment depending on the relative distribution of somatostatin in axon terminals. In those regions where the local somatostatin immunoreactivity was high, the sst2 receptors were preferentially intracellular and only a small portion of them was associated with plasma membranes. This distribution was proposed to be predictive of an efficient ligand-induced internalization of the receptors into the intracellular stores, which would indirectly imply that the sst receptors can traffic in-out plasma membranes. Interestingly, in these regions, the somatostatin-positive axon terminals were also positive for the sst2 receptors that would act as autoreceptors to control somatostatin release. The hypothesis was confirmed by a functional point of view [37] although also sst1 and sst3 receptors were proposed to have a main role in controlling somatostatin exocytosis [38].

Differently, in those regions where the somatostatin innervation was sparse, the sst2 receptors were predominantly inserted in plasma membranes, consistent with the conclusion that the peptide would diffuse over some distances to reach the receptors at concentration almost insufficient to cause their internalization (as observed in the regions with the dense innervation of the peptide). 

This mechanism of distribution is indicated with the term “volume diffusion” [39,40] and accounts for the ability of endogenous neurotransmitters/peptides to reach neuronal processes far from the site of release, through a mechanism(s) of distribution which mainly operates in the extracellular spaces and in the cerebrospinal fluid and that assures the progression of neuronal signalling from signal-secreting sender structures towards receptor-expressing targeted neurons and astrocytes located far away from the site of release. 

Such a mode of distribution was postulated to account for the propagation of most of the neuropeptides in the CNS [41] and depends on the different fate of the peptide and of the neurotransmitter(s) in the extracellular space. Once released in the synaptic cleft, the neurotransmitters (i.e., GABA) are usually rapidly taken-up because of the presence of the specific transporters in the surrounding cell membranes that end their neuromodulatory functions, also limiting their diffusion in the extracellular liquid. Differently, peptides (i.e., somatostatin) long last at the outer side of the neuronal plasma membranes because of the lack of an efficient uptake mechanism and can diffuse in the extracellular liquid, until soluble peptidases cleave and inactivate them. Somatostatin can therefore exert both autocrine and paracrine control mechanisms nearby the site of release (i.e., preferentially extrasynaptically), but also at distant brain targets (Figure 2), being pivotal for the diffusion and the amplification of the synaptic communication into brain regions vicinal to those where the first releasing activity occurs. The peptide amplifies the neuronal signalling, implementing the inter-neuronal communications between vicinal brain regions.

Studies were also dedicated to investigate whether and how somatostatin presynaptically controls the release of GABA but, to the best of our knowledge, the few available data do not support any definitive conclusion [42,43,44]. 

## 3. Somatostatin and Glutamate Presynaptically Modulate Each Other in the Central Nervous System

In a simplistic view, in most brain regions (i.e., the cortex, the hippocampus, the striatum) GABAergic interneurons make strict contacts with glutamatergic synapses and are functionally controlled by glutamate released by the presynaptic components of the chemical synapsis as well as by neighbouring structures (i.e., the astrocytes). Conversely, GABA inhibits glutamate exocytosis by activating presynaptic release-regulating GABA_B_ heteroreceptors on glutamatergic nerve terminals/axonal processes [35]. 

The role of somatostatin in this context has been long matter of discussion, since the data available in the literature unveiled both inhibitory and excitatory actions [44,45,46] of the peptide on glutamatergic transmission at least in hippocampal [47] and cortical tissue preparations [48]. The contrasting observations were ascribed to the involvement of both direct (involving either pre- or postsynaptic sst receptors) and indirect (involving intermediate neurotransmitters controlled by somatostatin) cascades of events. It was, however, in 1997 that Bohem and Betz definitively proved the inhibitory regulation that somatostatin exerts at central excitatory synapses via presynaptic receptors belonging to the SRIF1 receptor subfamily [49]. The existence of these presynaptic inhibitory release-regulating somatostatin heteroreceptors was soon after confirmed by Tallent and Siggins in 1997 [50] and their reciprocal distribution at both presynaptic and postsynaptic levels in the CNS was analysed in 2000 by Schulz and colleagues ([51], but see also [52]).

Then, in 2004, new insights were provided supporting the existence of presynaptic sst heteroreceptors controlling glutamate outflow. The results confirmed the existence of presynaptic, release-regulating sst receptors coupled to PTX-sensitive G proteins negatively linked to a AC/cAM/PKA-dependent pathway (Figure 3) ([53] but see also [54]). Grilli and colleagues [53] demonstrated that in cortical synaptosomes the K^+^-depolarization and the consequent Ca^2+^ influx through VOCCs doubles the synaptosomal content of cAMP, possibly by activating the AC subtype 1 (AC1) which is a Ca^2+^-dependent enzyme having a presynaptic localization. Somatostatin halved the glutamate exocytosis elicited by the K^+^ depolarization and nulled the glutamate overflow elicited by forskolin, which relies on intraterminal AC-sensitive pathway(s). Since AC1 is inhibited by sst receptors (see [55] for a review), it was postulated that the enzyme might represent the functional target through which the presynaptic sst receptors control the glutamate exocytosis.

Notably, the inhibitory effect was observed when synaptosomes were exposed to concentration of the peptide in the low nM range, consistent with the endogenous amount expected to be present in the CNS. Furthermore, by using selective sst1 to sst5 ligands, it was demonstrated that the sst2 subtype was the receptor involved in the presynaptic control [53].

Few years later, in an in vitro model of hippocampal hyperactivity, it was reported that somatostatin reduces the phosphorylation of the NMDA subunit NR1 [56], an event that was related to decreased postsynaptic excitatory currents due to the presynaptic somatostatin-mediated inhibition of excitatory transmission [49,50,57]. Soon after, evidence was provided showing that a very low (femtomolar) concentration of somatostatin also modulates presynaptic NMDA receptors in synaptosomes from rat brain cortex by reducing the calcium influx elicited by the ionotropic receptor [58]. 

Starting from these first observations, further evidence was published demonstrating that somatostatin inhibits presynaptic glutamate release in different brain regions [44,59,60,61,62]. The somatostatin-induced inhibition of glutamate release in the hippocampus was proposed to mainly involve the activation of sst2 receptors [63], although also the sst1 and sst4 receptor subtypes were proposed to participate to the presynaptic event [64]. 

The ability of somatostatin to control presynaptically the efficiency of glutamate transmission (particularly in the hippocampus and the cortex) well correlates with the physiological role of the peptide in the CNS and, in the case of maladaptive changes or disrupted peptidergic innervation, with the development of neurological disorders such as epilepsy, pain, stress and Alzheimer’s disease. In this regard, a large body of evidence support the involvement of presynaptic sst2 receptors in seizure susceptibility [65,66,67,68] also indicating these sst receptor subtypes as potential targets for novel therapeutic strategy to manage this disorder [18,63,65], but also in delaying the unavoidable deterioration of central functions that occurs during ageing [19].

When discussing the interaction between somatostatin and glutamate, it is, however, worth reminding that also glutamate controls somatostatin release in a reciprocal interaction which finely tunes the strength of the chemical synapsis. Evidence was provided supporting the existence of presynaptic NMDA heteroreceptors on somatostatinergic nerve endings. The receptors were unable to modify the release of the peptide in resting conditions, but significantly improved that elicited by a mild (high K^+^) depolarizing stimulus when activated by glutamate, glycine or D-serine alone to an extent that could not be further amplified when exposing synaptosomes concomitantly to glutamate and glycine (as usually observed with other NMDA receptors controlling the release of classic neurotransmitters) [69]. The main characteristics and the pharmacological profile of these receptors suggested that they belong to a particular class of NMDA receptors typified by low sensitivity to Mg^2+^ ions, high sensitivity to glycine and H^+^ and for which glutamate is not essential to trigger the functional responses. The main pharmacological features of this receptor has been recently described in a review dedicated to presynaptic NMDA receptors [58] and will be not further discussed here.

More recently, evidence was provided supporting the existence of presynaptic calcium-permeable kainate receptors containing GluR5/GluR6 subunits in somatostatinergic interneurons in the stratum radiatum of the hippocampus [70]. Differently, to the best of our knowledge, data supporting the existence of presynaptic release-regulating AMPA receptors and/or metabotropic glutamate receptors controlling somatostatin exocytosis are not available so far in the literature.

## 4. Somatostatin Receptors and Metamodulation in the Central Nervous System

When analysing the mechanisms of control of transmitter exocytosis, the impact of selected presynaptic receptors is usually analyzed individually, independently on whether these receptors colocalize with other ones that could be activated by the same neurotransmitter or by other modulators. The study of the presynaptic mechanisms of control of neurotransmitters release, however, unveiled the coexistence and even in some cases the physical association of auto- and heteroreceptors, whose concomitant activation modulates a common functional response (i.e., the exocytosis of neurotransmitter elicited by a mild depolarizing stimulus). This receptor-receptor cross-talk is at the basis of the “metamodulation” of synaptic transmission and accounts for the complexity of the mechanisms of control of neurotransmission in the CNS [71,72,73,74,75]. 

The term “metamodulation” [39,75,76] was first proposed to recapitulate the physical and functional integration of colocalized GPCRs [72,73,74,75], with particular emphasis to their impact on the efficiency of neurotransmitter release, but it is now extended also to the receptor-receptor interactions involving ionotropic receptors [77,78] or transporters [58]. 

Starting from the multiplicity and the complexity of the functional responses that originate from the metamodulation, it is worth reminding that, in most cases, each receptor participating to the receptor-receptor cross-talk exerts on its own a direct control of the neurotransmitter release that, however, is significantly modified when the colocalized receptor(s) is concomitantly activated. In other cases, one receptor can be *per se* functionally silent (i.e., unable to modify on its own the efficiency of neurotransmitter release) but it can modulate the colocalized receptor. It is the case of the CXCR4 receptor that colocalizes and functionally controls the NMDA receptor function in both glutamatergic and noradrenergic nerve endings [77] or of the sst5 heteroreceptor controlling the NMDA-mediated releasing activity in noradrenergic terminals [12]. 

Whatever the impact of the presynaptic receptors on the releasing activity, the final outcome of the metamodulation always differs from the mere sum (or difference) of the effects elicited by each receptor alone, unveiling the allosteric nature of the cross-talk. 

The functional interaction is referred to as an “agonist-like interaction” when the final effect is higher than the sum of the receptor-induced events (i.e., the mGlu1-mGlu5 receptors [76] or the nACh-NMDA receptors interactions [58]), or, conversely, it is defined an “antagonist-like interaction” if the outcome is quantitatively lower (i.e., the mGlu2/3 and the 5-HT_2A_ receptors [79]). 

Somatostatin receptors heterodimerize to form oligomeric structures typified by pharmacological properties different from those of the homomeric assemblies [80]. The heterodimerization of sst receptors was first proposed based on the observation that neurons express more than one sst receptor subtype [81]. The finding raised the question on whether the colocalized sst receptors are redundant or if they interact each other to improve/modify their functional outcomes. The results allowed to conclude that sst2 and sst4 receptors can functionally couple in both a cooperative and a competitive manner in the CA1 region of the hippocampus to control glutamatergic excitability [67,82]. Furthermore, in the same brain region functional interactions were also proposed to link sst1 and sst2 receptors [68] and sst1 and sst4 receptor subtypes [64]. Finally, sst5 receptor subtypes were reported to heterodimerize with sst1 but not with sst4 receptors, then indicating that heterodimerization is not a general phenomenon, but it is preferentially restricted to certain receptor subtypes combinations. The outcome of these receptor-receptor interactions, however, remains elusive, nor evidence have been provided supporting their existence in nerve terminals and synaptic boutons to metamodulate presynaptically neurotransmitter exocytosis. sst receptors were also proposed to oligomerize with GPCRs belonging to other receptor families, namely the opioid and the dopaminergic receptors [83,84,85], therefore increasing the complexity of their role in the central metamodulation. 

Finally, the term metamodulation can be also used to describe the receptor-receptor interactions involving ionotropic and metabotropic receptors [12,58], as described in the next paragraph that is dedicated to the metamodulation, bridging the sst5 and the NMDA receptors on noradrenergic nerve endings.

## 5. sst5 and NMDA Receptor-Receptor Interaction: A Model of Presynaptic Metamodulation

The mechanism of volume diffusion is particularly relevant for those neurons that do not make synaptic contact with other neuronal structures. It is the case of the noradrenergic projections ascending from the *Locus coeruleus* (LC) to upper brain regions. Most of the catecholaminergic varicosities along the axonal arborizations toward the innervated structures fail to make synaptic contact [40,86], consistent with the conclusion that noradrenaline released by these processes mediates a parasynaptic, rather than a synaptic, transmission that would involve high-affinity non-synaptic receptors. At the meantime, the noradrenergic varicosities/synaptic boutons preferentially sense the chemical signal from the surrounding network through neurotransmitters/peptides that diffuse into the extracellular space and establish a non-synaptic communication with the respective heteroreceptors located on the noradrenergic projections [40].

As far as the volume diffusion of somatostatin is concerned, the efficiency of the non-synaptic somatostatinergic signal is defined by the kinetics of peptide diffusion in the extracellular space, by the affinity of the ligand for the respective receptor(s) and by the pattern of expression and distribution of the receptors in the receiving neuronal processes as already discussed by Dutar and colleagues (see [37]). 

LC noradrenergic neurons and axonal processes are endowed with somatostatinergic receptors that can mediate the synaptic control of the peptide on the catecholaminergic innervation [87,88]. In particular, starting from the 2000, evidence was provided suggesting the existence of sst5 receptor subtypes on noradrenergic hippocampal synaptosomes (that could originate from either varicosity and synaptic terminals of the LC projecting neurons to the hippocampus [12]). Somatostatin acting at these presynaptic sst5 heteroreceptors failed to modify *per se* the spontaneous release of preloaded [^3^H]noradrenaline, but it hugely affected the releasing activity elicited by NMDA heteroreceptors in the same terminals [89]. At a first glance, the results seemed consistent with a positive allosteric GPCR-mediated metamodulation of presynaptic release-regulating NMDA receptors, as already described to occur for other GPCRs colocalized and positively coupled to NMDA heteroreceptors [58,90,91]. The novelty of the study, however, relied on the finding that activation of the sst5 receptors disclosed the release-regulating activity of the colocalized NMDA receptors also in the presence of physiological amount of [Mg^2+^]_out_, an ionic condition that should *per se* impede the NMDA-mediated releasing activity. Interestingly, the efficiency of somatostatin in disclosing the NMDA-mediated noradrenaline release also emerged in hippocampal slices, a more intact tissue preparation, ruling out the possibility that the “activation” of the NMDA-mediated effect was related to non-specific events due to the characteristics of the synaptosomal preparation [92].

The studies dedicated to investigate the molecular pathway(s) involved in the sst5-NMDA receptor-receptor interaction unveiled a complex scenario involving two main concomitant pathways. The first one relied on a Ca^2+^-dependent, calmodulin kinase II (Ca-CAMII)-mediated pathway that, possibly by phosphorylating the intraterminal sequences of the GluN subunits, reduces the sensitivity of the NMDA heteroreceptors to the [Mg^2+^]_out_ in resting conditions, then permitting its releasing activity also in physiological condition. The second one involved a PLC-induced, PKC-dependent, src-mediated phosphorylative pathway(s) which, by phosphorylating tyrosine residues of the GluN subunits, improved the gating of the voltage-dependent ionic channel associated to the NMDA receptor, amplifying the releasing efficiency of the presynaptic release-regulating NMDA heteroreceptors on noradrenergic nerve terminals (Figure 4). 

To the best of our knowledge, these studies provided the first evidence that a GPCR (i.e., the sst5 in our case) involved in a receptor-receptor functional cross-talk can both enable and amplify the release efficiency of colocalized NMDA receptors. The discrimination between the enabling activity and the amplification of the NMDA-mediated functions was made possible because of the main feature of the technique used (i.e., the “superfusion of a thin layer of synaptosomes [31,76]) which permits to measure the basal activity of presynaptic release-regulating receptors in a ligand-free condition, then allowing to discriminate between activation and potentiation of presynaptic release regulating receptors.

The aspect is particularly intriguing taking into consideration the need of drugs able to modulate the NMDA-mediated functions for therapeutic purposes. The sst5 ligands could represent a new class of indirect allosteric modulator of NMDA receptors, to be proposed to control and restore the central NMDA-mediated signalling in pathological conditions. 

Due to the pivotal roles of noradrenaline, somatostatin and glutamate in the processes of learning and memory, it was proposed that the sst5-mediated tuning of the NMDA heteroreceptors controlling noradrenaline release in hippocampus could be relevant to modulate the synaptic adaptation underlying learning and memory storage. Accordingly, the molecular cascade of events above described was found to account for the promnesic effect of CGP36742, a GABA_B_ receptors antagonist with high affinity for the GABA_B_ heteroreceptors on somatostatinergic terminals but weak activity towards the GABA_B_ autoreceptors and that is typified by strong nootropic activity in in vivo studies [92].

## 6. Conclusions and Future Perspective 

The review aimed at resuming most of the data concerning the role of somatostatin as presynaptic modulator of neuronal functions in the CNS of mammals, paying particular attention at the functional inter-relationship connecting somatostatin to the glutamatergic and the noradrenergic systems. It emphasizes some aspects of the neurobiology of this peptide that are relevant for the comprehension of its ability to control synaptic functioning. By discussing the role of somatostatin as a presynaptic modulator of glutamatergic and noradrenergic signal, we re-propose the main concepts of volume diffusion and metamodulation, which revolutionized the classic view of central chemical transmission. By dissecting the impact of somatostatin in these processes we tried to highlight the main role of the neuropeptide in controlling the synapsis efficiency, convinced that improving the knowledge of these molecular events is crucial for understanding the mechanisms of central plasticity, but it would also suggest new therapeutic approaches for the cure of central disorders. 

## Figures and Tables

**Figure 1 ijms-22-05864-f001:**
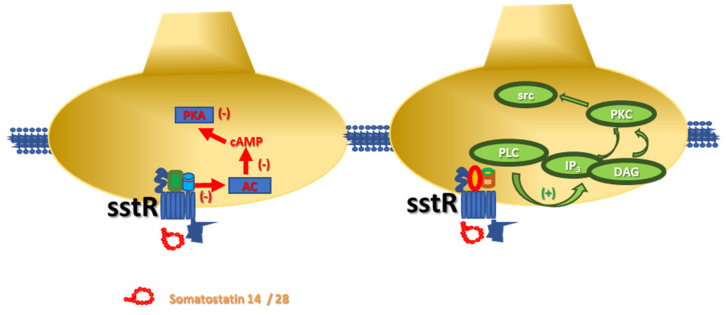
Schematic representation of the intraterminal enzymatic pathways coupled to sst receptors in isolated nerve endings. (**left**) Somatostatin-14/28 binds sst receptors negatively coupled to adenylyl cyclase (AC) and reduces the endogenous production of cyclic adenosyl monophosphate (cAMP) and the protein kinase A (PKA)-mediated phosphorylative pathways. (**right**) Somatostatin-14/28 also interacts with sst receptors positively associated to phospholipase C (PLC) then favouring its translocation to the synaptic membranes where the enzyme hydrolyses the phosphoinositides producing inositol trisphosphate (IP_3_) and diacylglycerol (DAG), which in turn activate protein kinase C (PKC) and cytosolic tyrosine kinases (src).

**Figure 2 ijms-22-05864-f002:**
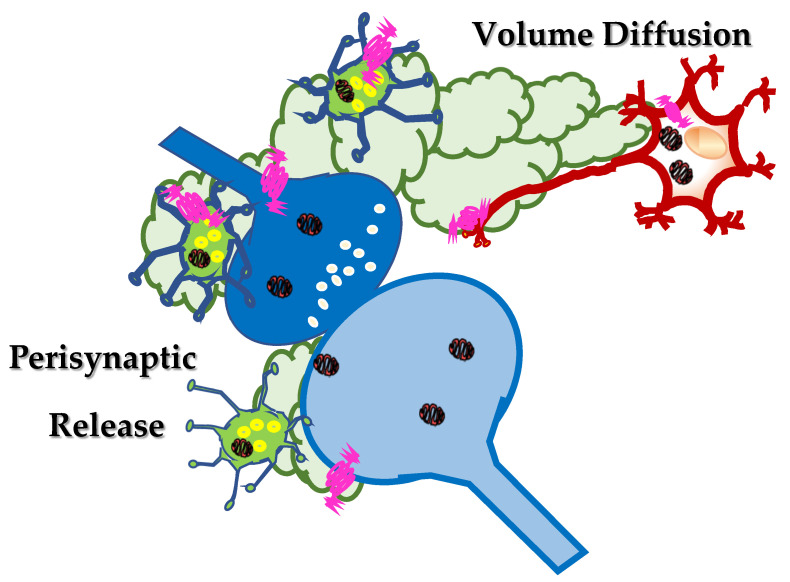
Somatostatin localizes in inter-neuronal networks (green neurons) and it is preferentially released, upon application of a depolarizing stimulus, perisynaptically, far from the synaptic active zone. Here, somatostatin can activate auto- and heteroreceptors located extrasynaptically, close to the synaptic active zone, that are referred to as perisynaptic receptors. Somatostatin can also diffuse in the extracellular space, throughout the mechanism of the “volume diffusion” (light green clouds), to activate distant sst receptors located on brain targets far from the site of release.

**Figure 3 ijms-22-05864-f003:**
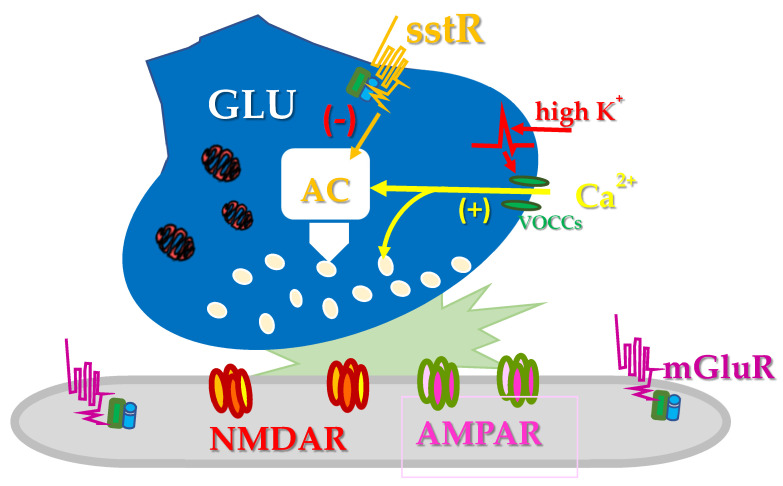
Depolarization with high-KCl-containing medium favours the opening of VOOCs and the influx of Ca^2+^ ions that triggers glutamate exocytosis either directly through both AC/cAMP/PKA-independent and AC/cAMP/PKA-dependent pathways. Presynaptic release-regulating sst2 heteroreceptors hamper the AC/cAMP/PKA-dependent pathway, then reducing glutamate exocytosis. The sst2-mediated inhibitory effect in turn reverberates on the functions of both metabotropic (mGlu) and ionotropic (NMDA and AMPA) glutamate receptors located postsynaptically, influencing the strength of the connection at glutamatergic synapses.

**Figure 4 ijms-22-05864-f004:**
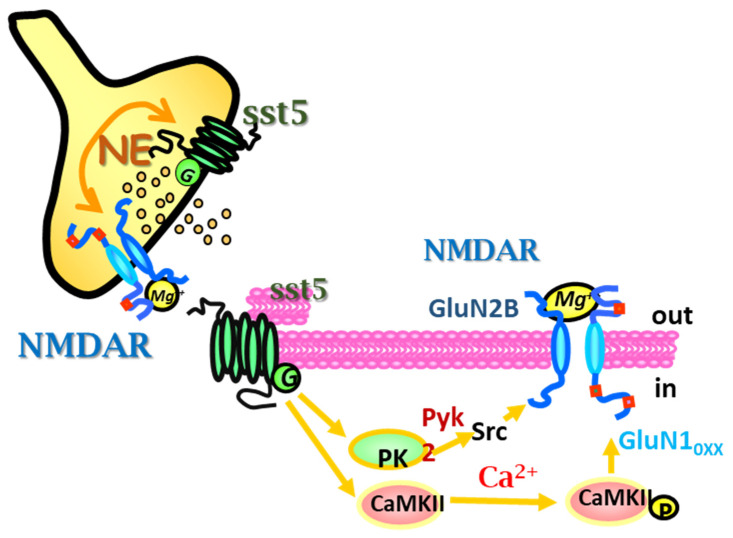
sst5 receptors activate and upregulate the colocalized and functionally coupled NMDA receptors in hippocampal isolated nerve terminals. A CaMKII-mediated pathway accounts for the sst5 receptor-mediated activation of NMDA receptors in the presence of physiological Mg^2+^ while a concomitant PLC/PKC/Src-mediated enzymatic pathway involving the proline-rich tyrosine kinase 2 (Pyk2) best accounts for the up-regulation of the NMDA receptor-mediated releasing activity.

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
