# Peer review of "Somatostatin, a Presynaptic Modulator of Glutamatergic Signal in the Central Nervous System"

_ijms, 2021, doi:10.3390/ijms22115864_

Round 1

Reviewer 1 Report

In the review “Somatostatin, a presynaptic modulator of glutamatergic signal in the central nervous system” the authors provide an overview of somatostatin impact on chemical synapses. After an introduction on the somatoninergic transmission, the authors focus on somatostatin action at the pre and post synaptic sites. In the last part of the rew, they introduce the concept of somatostatin-mediated metamodulation of synaptic transmission, anticipating an important role of this neuropeptide and its receptors in central plasticity and disorders.

A similar article was recently published in IJMS (Diversity and Function of Somatostatin-Expressing Interneurons in the Cerebral Cortex https://doi.org/10.3390/ijms20122952, not cited). In the submitted paper the authors expand the description on the mechanism of somatostatin action at the presynaptic site and introduce the idea of somatostatin-mediated metamodulation of synaptic transmission, therefore from this point of view the document is original

There are however some points that need to be fixed.

  • The review is interesting but not always clear. The mechanisms of reciprocal regulations in neurotransmission are certainly complex, but probably a remodulation of the sentence would better convey the authors’ idea (ie, lines 301-303; 424-433)
  • The part relative to metamodulation is an important and emerging topic in the field of neurotransmission and should be expanded.
  • The term “byophase” is misleading. “extracellular space” “extracellular liquid” would probably better convey the authors’ idea
  • The concept of “volume diffusion” is not so recognized worldwide. It is clearly explained in the text but should probably be removed or rephrased in the abstract.
  • Sst referring to a protein should be indicated by capital letters
  • 21+8 out of 99 articles cited come from the authors’ group. Although the authors’ contribution to the field is unquestionable, important works on somatostatin-and its involvement in neurotransmission are missing
  • Figure 2 caption: “somatostatin localizes in interneuronal networks (green particles)” Perhaps “green neurons” or “green cells” is more appropriate
  • Figure 2 cartoon: the yellow/green arrows to indicate somatostatin diffusion is not completely clear
  • Figure 3 caption “depolarization with high-KCl-containing ionic medium allows the influx of Ca2+ ions through the VOCCs and triggers glutamate exocytosis through activation of a AC/cAMP/PKA-dependent pathway”. Please, substitute with “dependent and independent pathways”. Indeed, there is also a direct Ca2+ effect on synaptic release.
  • Figure 3 cartoon: the yellow arrows incating ca2+ influx are not clear.
  • Figure 4 caption. Please define Pyk2, NE. What does “ongoing NMDA receptor activity” mean?
  • There are some typos in the text es 347 iyt, 380 missionthat etc

Author Response

We thank the referee for the positive comment to our work. 

Reviewer 2 Report

This is a reasonably well-written manuscript. Please check carefully and correct typo throughout the manuscript.

Several sentences are unclear. Please simplify sentences in lines 58 and 279. In line 330: "...impressive impact" - in what aspects it is impressive.

References required for statements in lines 267, 319, 383 and 429.

At the beginning, authors have identified two types of somatostatin (som 14 and som 28). Relevance of these two peptides in the function of somatostatin is not described. Do they have different affinity to ss receptors? 

Review is focused on modulation of memory and learning and several references deal with hippocampus. A description of neuronal sources of somatostatin in the hippocampus and the cerebral cortex would be relevant to the article (e.g., what type of neurone are they? what % of interneurons contain somatostatin?). 

Similarly, what is the density and spatial distribution of somatostatin receptors in the cerebral cortex and the hippocampus? 

In abstract, authors have indicated "located far away from the site of release" (line 17/18); it is not clear how far somatostatin diffuse in the brain; how long the biological activity of released somatostatin last in extra synaptic site; what mechanisms terminate the action of somatostatin/ destroy released somatostatin. 

Author Response

(The authors gave the same response as above.)

Round 2

Reviewer 2 Report

Recommend accepting in the current form. Would like to congratulate authors for a comprehensive review. Appreciate considering my suggestions.